# Syndromic Surveillance Systems for Mass Gatherings: A Scoping Review

**DOI:** 10.3390/ijerph19084673

**Published:** 2022-04-13

**Authors:** Eliot Spector, Yahan Zhang, Yi Guo, Sarah Bost, Xi Yang, Mattia Prosperi, Yonghui Wu, Hui Shao, Jiang Bian

**Affiliations:** 1Department of Health Outcomes and Biomedical Informatics, University of Florida, Gainesville, FL 32610, USA; eliotzbot@ufl.edu (E.S.); yiguo@ufl.edu (Y.G.); sarah.bost@ufl.edu (S.B.); alexgre@ufl.edu (X.Y.); yonghui.wu@ufl.edu (Y.W.); 2Department of Pharmaceutical Outcomes & Policy, University of Florida, Gainesville, FL 32610, USA; yahan.zhang@ufl.edu (Y.Z.); hui.shao@cop.ufl.edu (H.S.); 3Department of Epidemiology, University of Florida, Gainesville, FL 32610, USA; m.prosperi@ufl.edu

**Keywords:** syndromic surveillance, public health surveillance, mass gathering medicine, infectious disease surveillance, disaster management, emergency medicine

## Abstract

Syndromic surveillance involves the near-real-time collection of data from a potential multitude of sources to detect outbreaks of disease or adverse health events earlier than traditional forms of public health surveillance. The purpose of the present study is to elucidate the role of syndromic surveillance during mass gathering scenarios. In the present review, the use of syndromic surveillance for mass gathering scenarios is described, including characteristics such as methodologies of data collection and analysis, degree of preparation and collaboration, and the degree to which prior surveillance infrastructure is utilized. Nineteen publications were included for data extraction. The most common data source for the included syndromic surveillance systems was emergency departments, with first aid stations and event-based clinics also present. Data were often collected using custom reporting forms. While syndromic surveillance can potentially serve as a method of informing public health policy regarding specific mass gatherings based on the profile of syndromes ascertained, the present review does not indicate that this form of surveillance is a reliable method of detecting potentially critical public health events during mass gathering scenarios.

## 1. Introduction

As defined by the Centers for Disease Control and Prevention (CDC), public health surveillance is the “ongoing systematic collection, analysis, and interpretation of outcome-specific data for use in the planning, implementation, and evaluation of public health practice,” which has been instrumental in the reduction in mortality from exposure to infectious diseases and environmental toxins [1]. The arm of public health surveillance that deals specifically with the early detection of disease outbreaks or clusters of adverse health emergencies is referred to as syndromic surveillance and can be defined as “an investigational approach where health department staff, assisted by automated data acquisition and generation of statistical alerts, monitor disease indicators in real-time or near-real-time to detect outbreaks of disease earlier than would otherwise be possible with traditional public health methods” [2]. Syndromic surveillance involves the near-real-time collection and analysis of data from a multitude of sources, ranging from emergency departments to web queries to veterinary lab data [3,4,5]. A provisional diagnosis, or “syndrome,” can be established via the synthesis of clinical features, disease trends, and surrogate measures (e.g., derived from pharmaceutical sales and doctor visits, among others) [6]. Syndromic surveillance has been shown to be a potentially effective method for the early detection of seasonal outbreaks of influenza as well as other pathogens, and previous research has shown the viability of this surveillance strategy for detecting larger outbreaks (>1000 symptomatic cases) [7,8]. Additionally, syndromic surveillance is potentially useful in occupational scenarios, where emerging threats such as COVID-19 or influenza pose a risk to workplace staff [9,10].

Mass gatherings, such as large sporting events, often result in a heavy strain on regional healthcare systems due to a variety of factors, such as the influx of non-local travelers, reduced healthcare provider availability, communication hurdles, and increased non-endemic disease variation [11]. To monitor in real time and respond appropriately to public health events by virtue of mass gatherings, syndromic surveillance systems are often used [12]. Syndromic surveillance strategies have been utilized in a wide variety of mass gathering settings, including the London 2012 Olympic and Paralympic Games, the 8th Micronesian Games in 2014, and large camping events held by youth organizations [13,14,15,16]. Considering the widespread prevalence of COVID-19 and other infectious diseases across the globe, effective syndromic surveillance systems will be required to facilitate the safety of future large-scale social events [17].

While there is a small body of literature centered on the use of syndromic surveillance systems for the monitoring of health events arising from mass gatherings, there is no scoping review that details the specific methodologies and applications of these systems [6,18]. We aimed to conduct a thorough review of the technical aspects of existing syndromic surveillance technology to uncover important areas of improvement or challenges facing the implementation of these systems in a mass gathering scenario, as well as to increase understanding of the importance of syndromic surveillance for disease prevention and management related to mass gatherings. The primary characteristics under review are (1) the main methodologies of data collection and analysis inherent to syndromic surveillance systems as oriented towards mass gathering scenarios, (2) the degree of preparation and collaboration required to effectively operate syndromic surveillance systems for mass gathering scenarios, and (3) the degree to which prior surveillance infrastructure is used for the implementation of this surveillance strategy at mass gathering events.

## 2. Materials and Methods

This scoping review was conducted in accordance with the Preferred Reporting Items for Systematic Reviews and Meta-Analyses Extension for Scoping Reviews guidelines [19]. PubMed, Embase, and Web of Science were searched from 1 January 2000 to 10 January 2021. Two reviewers independently carried out the review and information extraction in each step, with disagreements resolved through discussions with a third team member and the remaining team, if appropriate.

Query terms were iteratively developed to identify relevant papers related to the use of syndromic surveillance specifically for mass gathering scenarios. To conduct a comprehensive search, terms were refined to include a wide spectrum of mass gathering settings, as well as publications describing the use of systems which fell under the definition of syndromic surveillance, but which did not specifically refer to such systems as “syndromic” (Appendix A).

To be included in the review, the studies must have described the use of syndromic surveillance for specific mass gathering scenarios (Appendix A). Non-English studies, as well as abstracts or posters from conferences were excluded. Studies that discussed the general use of a syndromic surveillance system for mass gatherings but did not focus on a specific mass gathering event, were excluded. There were no restrictions on either the location of the study or the population examined in the study. A study was included if it did not refer to the surveillance system as a “syndromic” surveillance system, but the system used real-time or near-real-time data collection methods based on broad categories of symptoms or signs for the purposes of health surveillance in relation to a mass gathering scenario. Mass gathering scenarios were defined as planned gatherings in which attendance exceeded more than 100 people. Publications from prior to the year 2000 were excluded.

After removing duplicate records, two reviewers used the Covidence tool to independently screen the title and abstract of each article for inclusion. Articles that did not meet the inclusion criteria were excluded. A third reviewer resolved the disagreements regarding study relevance. The full-text records of the remaining articles were further reviewed for relevance.

Data extraction was performed on the remaining studies using forms designed to collect information relevant to the goal of the review. The two reviewers extracted the data from the articles independently and the data extraction forms were routinely compared between the two reviewers to ensure that the extracted data were consistent and representative of the included studies. New data fields were added iteratively based on information that was frequently present in the included studies. Consensus was reached for each article through discussions with the study team.

Data fields included were the location of the mass gathering event, mass gathering type (e.g., sporting event, music festival, etc.), dates of the gathering and surveillance period, number of attendees, whether the surveillance system was built for the event or previously in place (or modified, if applicable), the syndromes covered by the system, the data sources used by the system, the general data pipeline, temporality of data processing (e.g., real-time or near-real-time), preparation required for the implementation of the surveillance system, the process of data collection or the data processing pipeline, the software used, algorithm used, and highlights in the discussion on limitations of the systems. Some fields were excluded from the final analysis due to incomplete or missing data in many of the studies. For example, “software used” was excluded because information about specific software packages or programs was only explicitly described in a small proportion of studies. In some studies, it was apparent that software was used in some capacity; however, without being able to report specific information in the “software used” data field, the study could be misrepresented.

## 3. Results

Our search yielded 538 articles from the three databases (i.e., PubMed, Embase, and Web of Science) and two from other resources (cited in publications that underwent full-text review), of which 422 remained after deduplication (Figure 1).

Appendix A outlines the main characteristics of the systems detailed in each study. For studies in which the surveillance strategy was multifaceted and had many branches of surveillance activities (and in different subsystems), only the core components of the system were detailed. For example, for Super Bowl XLIX in 2016, multiple systems were outlined as part of the overall surveillance strategy, including hospital and urgent care surveillance, emergency room syndromic surveillance, and real-time onsite syndromic surveillance [20].

### 3.1. Characteristics of Included Studies

#### 3.1.1. Data Sources and Event Types

Of the 19 studies included for data extraction, 53% (n = 10) specifically indicated the use of data from hospital emergency departments for the syndromic surveillance system; 32% of the studies (n = 6) detailed the use of first aid stations and temporary or mobile clinics; 21% (n = 4) of the studies outlined the use of hospitals or care sites as data sources but did not indicate specific departments. One study examined the use of a mobile phone app as a data source. Veterinary clinics served as a data source for potential animal-borne pathogens in one study. Poison control and coroner investigation data were also used as data feeds in one study. One study used hospital outpatient data. Of the 19 studies identified, 42% (n = 8) used multiple types of care sites and departments (e.g., hospital EDs, event-specific mobile clinics, etc.) as data sources. There was a variety of event types in which syndromic surveillance was utilized, including sporting (n = 9), religious (n = 5), political or cultural (n = 4), and one natural (n = 1; 2017 solar eclipse viewings). This information is presented in Table 1 and Table 2.

#### 3.1.2. Data Types and Processing Procedures

Out of the 19 studies, 47% (n = 8) detailed the use of standardized reporting forms or surveys that healthcare staff, such as nurses, volunteers, and physicians, used to record basic demographic and syndrome data about a patient visit. This was most common in mobile clinic or first aid settings, where a centralized record-keeping infrastructure was not present. However, one study used self-reports in which patients detailed their own health condition and other factors, such as location and social encounter information, via a phone app [28]. The surveys or forms were administered over two primary mediums: paper-based, and digital offered via tablet or computer [12,20,21,22,23,24,25].

Forty-two percent (n = 8) of the studies detailed the systems in which alerts were produced automatically based on syndrome counts. These alerts were most often (n = 6) manually assessed by trained epidemiologists or public health officials to deem whether a further investigation was warranted. In one study, even though there was no automated alert system in place, some cases of syndromes, such as watery diarrhea and acute fever and neurological symptoms (AFNs), were investigated immediately because of potentially significant public health impacts [37]. Data were most often reviewed daily, either automatically or by manual analysis, and summarized (n = 14). In some systems (n = 3), data were processed in real time and through automated statistical analysis, indicating a high degree of temporal specificity regarding the presentation of syndromes in relation to a mass gathering event. Two studies reviewed the data retrospectively, which prevented the researchers or public health officials from investigating health events contemporaneously. While real-time or near-real-time analysis is a fundamental component of syndromic surveillance, these studies were included because they still implemented syndromic classification of healthcare data in relation to mass gathering events, examining the viability of this surveillance strategy in such scenarios [30,35].

#### 3.1.3. Preparation Steps

All studies detailed, to some degree, a preparation step that was necessary to ensure the reliability of the surveillance activities and processes during the actual mass gathering event. For example, training, or instructions, on how to use surveillance-oriented reporting mechanisms was explicitly mentioned in 12 studies. For the 2014 Arbaeenia Mass Gathering, an extensive training scheme was outlined in which surveillance staff were evaluated on a series of training data to ensure the proper administration and completion of surveys [27]. Twenty-six percent of the studies (n = 5) indicated the collection of baseline data prior to the mass gathering event. In all extracted studies, there was correspondence between at least one local, regional, or national public health agency and a care site or mobile clinic. Forty-seven percent (n = 9) of the studies detailed systems which previously existed but were enhanced, in some capacity, to meet the needs of the relevant mass gathering scenario. One of the robust pre-existing syndromic surveillance systems outlined was that of the Acute Care Enhanced Surveillance (ACES) system in Ontario, Canada, covering 184 acute care hospitals across the region and capturing data on 84 distinct syndromes [27].

#### 3.1.4. Syndromic Profiles

The scope and number of syndromic indicators or syndrome classifications ranged widely among the studies. Some studies limited the syndromes included in surveillance to a specific geographic region. For example, researchers conducted a system and disease risk assessment including a literature review of the disease patterns, assessment of disease databases, assessment of the public health and surveillance teams, and interviews and focus groups with key stakeholders within the Solomon Islands, where the mass gathering event was going to happen [37]. In another case, a system included a case definition for acute diarrhea as a response to a local outbreak of rotavirus that was occurring in South Tarawa at the time of the 2013 Kiribati Independence Celebrations [32]. Some of the systems (n = 5) used event-related tags to establish a clear connection between a patient visit and the mass gathering event. For the 2015 Special Olympics, proactive patient tagging was used to flag attendees of the event in ED clinical notes, which allowed subsequent algorithms to easily identify those patients and potential health concerns related to the event [21]. For surveillance activities surrounding the 2017 solar eclipse viewing events in Kentucky, US, “eclipse” was included as a unique syndromic classification [26]. “Inauguration-related” was included as a syndrome definition for surveillance activities oriented around the 2017 US presidential inauguration [25].

## 4. Discussion

In this scoping review, syndromic surveillance was examined as a potential strategy for providing support to and maintaining the robustness of public health infrastructure surrounding mass gathering scenarios. Nineteen studies were identified for data extraction and key information was collected about the mass gathering events and related syndromic surveillance activities. The systems described were implemented in a wide range of geographic and socio-economic settings, indicating a diverse potential for syndromic surveillance to meet the needs of different environments.

As identified in the present review, a core component of a reliable syndromic surveillance system or infrastructure, with regard to implementation during a mass gathering scenario, is ample preparation. In all studies examined, syndromic surveillance activities were preceded to some degree by a preparation phase. Since event-oriented surveillance is dependent on the integration of data from multiple sources as well as different mediums (e.g., hospitals and first aid stations), it is necessary that sufficient time be given to organize and facilitate collaborations among several entities. For the 2015 Super Bowl, communication was described among a large number of health agencies, healthcare providers, and businesses, including the Maricopa Department of Health, Maricopa County hospitals, local hotels, Arizona Department of Health Services and the U.S. Centers for Disease Control and Prevention, as well as other agencies and services. A vital component of the preparation phase prior to syndromic surveillance is training key staff members and physicians. This was especially important in scenarios where surveillance activities stemmed from event-based care sites, such as first aid stations and mobile clinics. There is a clear benefit to assessing disease profiles of regions proximal to the mass gathering event. In one circumstance, a literature review was conducted to identify disease trends within the area of operation as well as an assessment of pre-existing public health and lab surveillance infrastructure. This study also describes a risk assessment examining the characteristics of the mass gathering scenario [37].

Previous research has examined the potential for syndromic surveillance to detect disease outbreaks earlier than other forms of public health surveillance [7,38]. In the present review, the evidence to support these claims is limited with regard to a mass gathering scenario. While most of the studies indicated that the purported systems functioned reliably under the strain of increased clinical visits, none of the studies indicated that any significant outbreaks of disease were detected. In one case, a system was modified to meet the needs of an outbreak spurred by a mass gathering event [32]. Additionally, very few studies outlined any attempts at contemporaneously investigating or validating the data collected and analyzed by the system. In some cases, syndromes with low prevalence were suspected to not generate a sufficiently strong signal in the detection measures implemented by a system. However, without cross-validation with other data sources independent of the described system, it is difficult to establish whether the signals generated should warrant a potentially costly public health response.

There were some limitations concerning the present review. Due to the high variability in the design among the studies, it was difficult to extract standardized information from each study for tabulation in a data extraction form. Some of the studies did not include a high degree of specificity and details about the systems. Additionally, a thorough examination of the health outcomes measured by each syndromic surveillance system, in association with the described mass gathering events, has not been attempted.

## 5. Conclusions

Further research is necessary to conclude whether syndromic surveillance has a verifiable impact on the health of the populations under surveillance during mass gathering scenarios. The goal of the present review was to outline the process of preparing for and implementing this surveillance strategy at mass gathering events; however, further analysis of health and disease profiles at these events is necessary.

## Figures and Tables

**Figure 1 ijerph-19-04673-f001:**
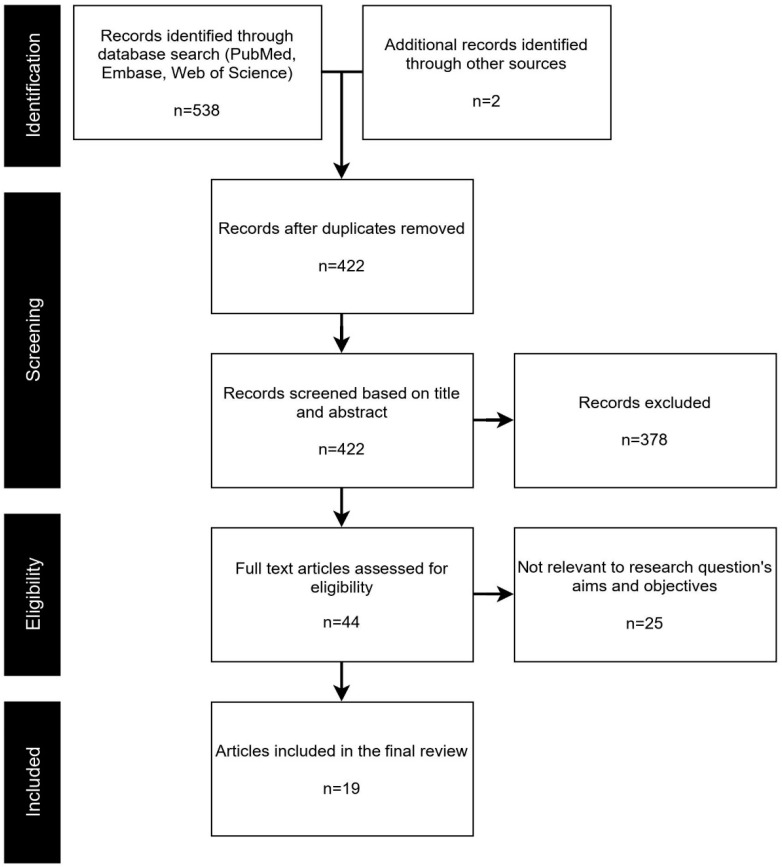
PRISMA study selection flowchart.

**Table 1 ijerph-19-04673-t001:** Mass gatherings.

Study	Gathering Type	Location	Main Event
White et al. (2018) [14]	Sporting	Pohnpei State, FSM	8th Micronesian Games
Ayala et al. (2016) [20]	Sporting	Maricopa County, Arizona, US	Super Bowl XLIX
Kajita et al. (2017) [21]	Sporting	Los Angeles, California, US	2015 Special Olympics
Bieh et al. (2020) [22]	Religious	Mecca, Saudi Arabia	2019 Hajj
Carrico et al. (2005) [23]	Sporting	Louisville, Kentucky, US	2002 Kentucky Derby
Elias et al. (2020) [24]	Religious	Maputo City, Mozambique	2019 Pope Francis Visit
Cherry et al. (2019) [25]	Political	Washington, D.C., US	2017 Presidential Inauguration
Heitzinger et al. (2020) [26]	Natural	Hopkinsville, Kentucky, US	2017 Solar Eclipse
Lami et al. (2019) [27]	Religious	Wassit Governate, Iraq	2014 Arbaeenia
Neto et al. (2020) [28]	Sporting	Rio de Janeiro, Brazil	2016 Summer Olympics
Elliot et al. (2012) [29]	Sporting	London, UK	2012 Summer Olympics
Sokhna et al. (2020) [30]	Religious	Touba, Senegal	2016 Grand Magal of Touba
Lami et al. (2019) [31]	Religious	Najaf/Karbala, Iraq	2016 Arbaeenia
Tabunga et al. (2014) [32]	Cultural	South Tarawa, Kiribati	2013 Kiribati Independence Day
White et al. (2017) [33]	Political	Apia, Samo	Third UN Conference on SIDS
Van Dijk et al. (2017) [34]	Sporting	Toronto, Ontario, Canada	17th Pan American and Parapan American Games
Todkill et al. (2016) [35]	Sporting	London, UK	2012 Summer Olympics
Muscatello et al. (2005) [36]	Sporting	New South Wales, Australia	2003 Rugby World Cup
Hoy et al. (2016) [37]	Cultural	Solomon Islands, Oceania	11th Festival of Pacific Arts

**Table 2 ijerph-19-04673-t002:** Data sources and processing procedures.

Data Sources	Publication	Processing Procedure	Data Reviewed
Hospital Emergency Department	Ayala et al. (2016) [20]	Automated search/aberration detection of clinical notes for event-related terms (Biosense)	Daily
	Kajita et al. (2017) [21]	Automated search/aberration detection of clinical notes (patients proactively tagged with event name; EARS)	Daily
	Bieh et al. (2020) [22]	Aberration detection based on ICD-10 diagnosis data	Hourly
	Carrico et al. (2005) [23]	Manual review of ED syndromic data by health department staff	Daily
	Cherry et al. (2019) [25]	Automated aberration detection based on chief complaint text data (ANCR-ESSENCE)	Daily
	Heitzinger et al. (2020) [26]	Automated aberration detection based on chief complaint data (ESSENCE)	Every 6 h
	Elliot et al. (2012) [29]	Automated monitoring of ED information system data	Daily
	Van Dijk et al. (2017) [34]	Automated aberration detection based on chief complaint data (ACES)	In real time
	Todkill et al. (2016) [35]	Automated aberration detection based on clinical diagnosis (EDSSS)	Retrospectively
	Muscatello et Al. (2005) [36]	Automated statistical analysis based on demographics, chief complaint and diagnosis codes	Daily
First Aid Stations or Event-Based Clinics	Elias et al. (2020) [24]	Software-based statistical analysis of tablet-based survey data (manually entered)	Daily
	Lami et al. (2019) [27]	Statistical analysis of form-based patient data (manually entered)	Daily
	Sokhna et al. (2020) [30]	Statistical analysis of free-text form-based patient data (manually entered)	Retrospectively
	Lami et al. (2019) [31]	Manual review of digital survey-based syndromic data by surveillance team	Daily
Mobile phone app	Neto et al. (2020) [28]	Automated analysis of user-submitted symptom and encounter data	In real time
Hospitals/Community Clinics	Tabunga et al. (2014) [32]	Manual review of staff-reported case presentations	Daily
Hospitals/Community Clinics/Game Venues	White et al. (2018) [14]	Data summaries produced with SAGES-OE based on surveillance-form collecting encounter/syndrome data	Daily
Sentinel sites	White et al. (2017) [33]	Data summaries produced with spreadsheet software based on surveillance register collecting encounter data	Daily
Sentinel sites/Public, Private and Temporary clinics	Hoy et al. (2016) [37]	Web-based database used to produce data summaries/reports based on reporting form used at sentinel sites	Daily

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
