# Peer review of "Syndromic Surveillance Systems for Mass Gatherings: A Scoping Review"

_ijerph, 2022, doi:10.3390/ijerph19084673_

Round 1

Reviewer 1 Report

Syndromic surveillance is a new technique that can provide interesting perspectives in public health. The authors make a useful summary of the literature.

  1. In the introductory part, lines 51-52, they note that this technique has been applied in a multitude of sources, ranging from emergency departments to web queries to veterinary lab data. They should recall here the use of data from workers' health surveillance that is carried out in the workplace. An example is reported in The Lancet Chirico F, Magnavita N. The West Nile virus epidemic-occupational insight. Lancet. 2019 Mar 30;393(10178):1298. doi: 10.1016/S0140-6736(19)30028-5].
    Sim JXY, Conceicao EP, Wee LE, Aung MK, Wei Seow SY, Yang Teo RC, Goh JQ, Ting Yeo DW, Jyhhan Kuo B, Lim JW, Gan WH, Ling ML, Venkatachalam I. Utilizing the electronic health records to create a syndromic staff surveillance system during the COVID-19 outbreak. Am J Infect Control. 2021 Jun;49(6):685-689. doi: 10.1016/j.ajic.2020.11.003. Bordonaro SF, McGillicuddy DC, Pompei F, Burmistrov D, Harding C, Sanchez LD. Human temperatures for syndromic surveillance in the emergency department: data from the autumn wave of the 2009 swine flu (H1N1) pandemic and a seasonal influenza outbreak. BMC Emerg Med. 2016 Mar 9;16:16. doi: 10.1186/s12873-016-0080-7.
    The authors can point to this or other studies as they please, but this point is important.
  2. the references have not been reported according to the editorial style of IJERPH

Author Response

Dear reviewer,

Thank you for the opportunity to resubmit the above-mentioned manuscript. We have revised the manuscript in response to your comments. We list each comment, followed by our responses.

  1. In the introductory part, lines 51-52, they note that this technique has been applied in a multitude of sources, ranging from emergency departments to web queries to veterinary lab data. They should recall here the use of data from workers' health surveillance that is carried out in the workplace. An example is reported in The Lancet Chirico F, Magnavita N. The West Nile virus epidemic-occupational insight. Lancet. 2019 Mar 30;393(10178):1298. doi: 10.1016/S0140-6736(19)30028-5].Sim JXY, Conceicao EP, Wee LE, Aung MK, Wei Seow SY, Yang Teo RC, Goh JQ, Ting Yeo DW, Jyhhan Kuo B, Lim JW, Gan WH, Ling ML, Venkatachalam I. Utilizing the electronic health records to create a syndromic staff surveillance system during the COVID-19 outbreak. Am J Infect Control. 2021 Jun;49(6):685-689. doi: 10.1016/j.ajic.2020.11.003. Bordonaro SF, McGillicuddy DC, Pompei F, Burmistrov D, Harding C, Sanchez LD. Human temperatures for syndromic surveillance in the emergency department: data from the autumn wave of the 2009 swine flu (H1N1) pandemic and a seasonal influenza outbreak. BMC Emerg Med. 2016 Mar 9;16:16. doi: 10.1186/s12873-016-0080-7. The authors can point to this or other studies as they please, but this point is important.

Response: Thank you for suggesting the inclusion of this research. We have added references to two of the aforementioned studies in lines 58-60 of the introduction.

  1. the references have not been reported according to the editorial style of IJERPH

Response: The citations have been updated to match the style of IJERPH (Multidisciplinary Digital Publishing Institute in Zotero).

Author Response

Dear reviewer,

Thank you for the opportunity to resubmit the above-mentioned manuscript. We have revised the manuscript in response to your comments. We list each comment, followed by our responses.

  1. My first and foremost remark is concerned with how you name your article. You call it a review. But for a full-scale review your analysis is obviously insufficient. A typical review usually includes systematising from one hundred to two hundred or even more (sometimes substantially more) literature sources. You may choose either of two ways, to “rebrand” your paper as a mini-review, or add / analyse much more literature. I am sure that the former way is preferable. In such a case please indicate it clearly in the Introduction that this is a mini-review and as it is, it does not necessarily cover all the literature published in the period you study

Response: We respectfully disagree, as our paper follows the scoping review protocol according to the PRISMA-ScR statement. Since we followed a systematic-scoping protocol to conduct the literature search, extract information from the records, and resolve disagreements among reviewers, we believe that it is appropriate to label the review as ‘scoping’ (as per PRISMA guidelines). The number of articles retrieved is reflective of the amount of research published on this topic at the time the literature search was conducted.

  1. It was unclear to me which algorithm you used for choosing the final nineteen works you farther studied. In Results section you note that initially you collected 422 publications. Nineteen is definitely less. As I understood, you isolated some “basic scenarios”, as you write “To be included in the review, the studies must have described the use of syndromic surveillance for specific mass gathering scenarios.” But please specify how you started to tackle the task of finding these “basic scenarios”. It may be done in Methods section.

Response: Thank you for this comment. We added sentences (Lines 108-109) to the Methods section that define ‘mass gathering scenarios’ as well as publication dates required for inclusion. Additionally, we added inclusion/exclusion criteria in an appendix. We also added a line (Lines 112-113) that indicates that the inclusion/exclusion criteria was used for screening the articles.

  1. In Supplementary Material, Table S1 must be presented as a real table or the title “Table” should not be used, as it is misleading now. After the “Table” indication, there should not be ordinary paragraphs of the text.

Response: Thank you for this suggestion. We have changed the title of ‘Table S1’ to ‘Appendix S1’.

  1. I could not see Table S2 in full, as there was overlapping of material. I tried to do it on two PCs and the result was the same. Please check it.

Response: We have decided that it would be best to include the finalized extraction form as a supplementary excel file, provided that is allowed. Table S2 has been removed from the  ‘supplementary’ word document.

  1. IJERPH adheres to numbering style of citing literature ([1], [2], etc.). Therefore, all the places where you write Name (Year), must be redone. E.g., on p. 5 you give the references as Hoy et al. (2016) and Tabunga et al. (2014). Please correct all such places, including Table 1 and Table 2. In columns “Study” and “Publication” please provide only consecutive numbers: [1], [2], …

Response: We have changed the citation style to match IJERPH standards. We included the citation number in brackets, as per guidelines, however it was not possible to make them consecutively numbered without changing the order of the references within the text.

  1. I would not write “5.Conclusions” as a subtitle on p. 7. You use Results / Discussion pattern, therefore there is no need in a separate title or subtitle “Conclusions”. If you remove it, the meaning of the last paragraph will be the same.

Response: Thank you for this suggestion. We have removed the subheader ‘Conclusions’ from this section.

  1. I strongly advise to reconsider the phrase “Syndromic surveillance strategies have been utilized in a wide variety of 64 mass gathering settings, including…” (2nd paragraph on p. 2). There is a semantic error in it. One cannot simply put planned events and unplanned events beside. The 2015 Paris terrorist attacks (to be more exact, they were Paris and Bruxelles attacks) cannot stand near such planned sporting events as Olympic Games! So, please remove these attacks from the list, lest it should make a great confusion among the readers.

Response: Thank you for clarifying this. The reference to the 2015 Paris terrorist attacks has been removed.

Round 2

Reviewer 2 Report

Everything is good now. I recommend the paper for publishing. Thank you.